# Akt, IL-4, and STAT Proteins Play Distinct Roles in Prostaglandin Production in Human Follicular Dendritic Cell-like Cells

**DOI:** 10.3390/ijms242316692

**Published:** 2023-11-24

**Authors:** Jihye Jeong, Jongseon Choe

**Affiliations:** Interdisciplinary Graduate Program in BIT Medical Convergence, Department of Microbiology and Immunology, School of Medicine, Kangwon National University, Chuncheon 24341, Republic of Korea

**Keywords:** follicular dendritic cell, Akt, prostaglandin, IL-4

## Abstract

This study aimed to explore the role of Akt protein in the induction and inhibition of prostaglandin (PG) in human follicular dendritic cell (FDC)-like cells. FDC-like cells and B cells were isolated from human tonsils. PG production was assessed using enzyme immunoassay, while the upstream cyclooxygenase-2 (COX-2) protein levels were measured using immunoblotting with FDC-like cells transfected with Akt siRNA to analyze the impact of Akt knockdown. The COX-2 expression and PG production induced with IL-1β were significantly increased by Akt knockdown. However, IL-1β did not significantly alter either total or phosphorylated Akt protein levels. Akt knockdown resulted in the augmentation of COX-2 expression induced by B cells, although the addition of B cells did not significantly modulate both total and phosphorylated Akt proteins. In contrast, IL-4 specifically exhibited a potent inhibitory effect on COX-2 protein induction and PG production via STAT6. The inhibitory activity of IL-4 was not hampered by Akt knockdown. Interestingly, COX-2 expression levels induced with IL-1β were markedly modulated with STAT1 and STAT3 knockdown. STAT1 silencing resulted in further augmentation of COX-2, whereas STAT3 silencing prohibited IL-1β from stimulating COX-2 expression. The current results suggest that Akt, IL-4, and STAT1 play inhibitory roles in PG production in FDC-like cells and expand our knowledge of the immune inflammatory milieu.

## 1. Introduction

Follicular dendritic cells (FDCs) constitute the essential stromal component of secondary lymphoid follicles [1]. FDCs are reported to be a type of fibroblast specialized in the retention and presentation of native antigens, promoting survival, proliferation, and differentiation of B cells [2]. They achieve these functions in coordinated interactions with B cells via direct cellular contacts and soluble factors. Our knowledge of some of FDC functions was obtained at the molecular level. For instance, Fcγ receptors and complement receptor CD21 are essential for immune complex retention of FDCs [3,4,5]. Chemokines, including CXCL13, are liberated from FDCs and crucially regulate B cell trafficking to the follicles [6]. However, the FDC molecules involved in follicular B cell proliferation and differentiation are poorly defined. To dissect the interaction mechanisms between FDCs and B cells at the molecular level, several laboratories, including ours, have established in vitro experimental models due to the technical difficulty of obtaining FDCs in sufficient numbers [7]. We have been preparing FDC-like cells from human tonsils as described before [8]. These FDC-like cells are primary fibroblasts displaying several similarities to FDC. They exhibit dendritic morphology and are negative for CD45, a hematopoietic stem cell origin antigen, but positive for prostaglandin I synthase (PGIS), an FDC antigen [9]. FDC-like cells enhance the survival and differentiation of B cells [8,10]. With the results procured with FDC-like cells, we have paid attention to cyclooxygenase-2 (COX-2) and prostaglandins (PGs) as potential mediators between FDC and B cells. PGs are derived from arachidonic acids after the sequential enzymatic reactions of COX-2 and respective prostaglandin synthases. The FDC is the major cell type expressing COX-2 in the germinal center (GC) of secondary follicles [11]. COX-2 is induced in FDC-like cells by various stimuli, including LPS, TGF-β, IFN-γ, PGs, and TNF-α [9]. COX-2 is also upregulated in FDC-like cells after coculture with B cells [11]. In contrast, COX-2 induction in FDC-like cells is inhibited by IL-4 [12]. PGs released from FDC-like cells promote survival, antigen-presenting capabilities, and differentiation of GC B cells [10]. The immune-stimulating function of PG was demonstrated in vivo since PG administration resulted in a marked increase in antigen-specific antibody production in mice [13]. Bernard et al. demonstrated an essential role for COX-2 in regulating humoral immunity [14], while Kojima et al. reported that PGs produced from nonhematopoietic mesenchymal stromal cells play a critical role in humoral immune responses in vivo [15]. Taken together, these results from our laboratory and others suggest that PGs liberated from stromal cells, such as FDCs, are important for normal humoral immunity.

This study was undertaken to determine the role of Akt, also known as protein kinase B, in both the induction and inhibition of COX-2 expression and the resultant PG production in FDC-like cells. Akt has been extensively studied in the field of biology and medicine [16]. However, the roles of this protein in the GC are largely unknown. IL-1β was chosen as the stimulus of COX-2 in this study since it exhibited a potent COX-2-inducing activity alone or in combination with other triggers. The potential role of Akt in COX-2 expression was further analyzed after coculture of FDC-like cells with B cells. The effect of Akt was also studied on the inhibitory activity of IL-4 in IL-1β-stimulated COX-2 expression. Although time-consuming, the immunoblotting technique was used in the current study. This methodology is highly specific to the final product of gene expression and allows us to differentiate phosphorylated Akt proteins from unphosphorylated ones. The effects of six STAT proteins in this process were also investigated. The current results reveal novel findings of the COX-2 induction in FDC-like cells and further our understanding of the regulatory functions of several proteins.

## 2. Results

### 2.1. Akt Knockdown Leads to Augmentation of IL-1β-Induced COX-2 Expression

To elucidate the intracellular molecular mechanisms underlying COX-2 induction in FDC-like cells, we looked into the role of Akt in this process. We have recently reported that Akt may play an inhibitory function in COX-2 expression induced by PGE_2_ [17]. This result prompted us to explore the potential role of Akt in COX-2 induction with IL-1β by comparing COX-2 expression levels between Akt-knockdowned cells and control cells. Successful knockdown of Akt proteins was substantiated with immunoblotting, as indicated by the average 82% (±8.8, *n* = 5) reduction in the Akt level (Figure 1A). In contrast to the marked reduction in total Akt proteins, Akt knockdown did not significantly diminish the expression levels of phosphorylated Akt (Appendix A). It appears that the remaining Akt proteins after Akt knockdown are just enough to exhibit the constitutively phosphorylated levels in these experiments. Phosphorylated Akt proteins are readily detected in unstimulated FDC-like cells [17]. Akt knockdown itself did not give rise to modulation of background level expression of COX-2. Since Akt displayed an inhibitory activity in a previous report, suboptimal doses of IL-1β were added to the cultures. IL-1β added at 25 or 50 pg/mL significantly increased COX-2 protein expression in control cells, which was further augmented in cells treated with Akt siRNA (Figure 1A,B). The augmenting effect of Akt silencing was also observed in the actual production of 6-keto-PGF_1α_ (Figure 1C), suggesting that the Akt molecule plays an inhibitory role in IL-1β-induced COX-2 protein expression and resultant PG production. 6-Keto-PGF_1α_ is a stable conversion product of PGI_2_ that is produced with PGI synthase (PGIS), a constitutively expressed enzyme in FDC-like cells. Neither IL-1β stimulation nor Akt knockdown modulated the expression levels of COX-1. These results led us to examine whether IL-1β would suppress the Akt activity by measuring Akt phosphorylation levels. Consistent with our previous results [17], phosphorylated Akt proteins were readily detected in unstimulated cells. In time-course experiments, Akt phosphorylation levels were measured up to 120 min poststimulation with IL-1β. As shown in Figure 1D, the presence of IL-1β did not suppress Akt phosphorylation but gave rise to a slight but nonsignificant augmentation of phosphorylated Akt proteins at 30 and 60 min.

### 2.2. The Inhibitory Effect of Akt Was Also Observed in COX-2 Induction after Coculture with B Cells

Since COX-2 protein expression in FDC-like cells was induced with activated B cells, as described in our recent report [11], we looked into the role of Akt in this phenomenon. The coculture of FDC-like cells was carried out with tonsillar B cells. B cells were freshly isolated from tonsils with a purity of more than 90% of CD3^-^CD20^+^ phenotype, as shown in a representative result (Figure 2A). B cells were a mixed population containing CD20^hi^CD38^hi^ GC B cells and CD20^lo^CD38^lo^ naïve B cells. FDC-like cells were harvested at the end of the 24-h culture after removing contaminating B cells with gentle pipette washing, followed by immunoblotting to measure the expression levels of COX-2 proteins. Compared to FDC-like cells, Akt proteins in B cells were nonphosphorylated (Figure 2B). Although the addition of B cells did not lead to significant modulation of Akt phosphorylation (Figure 2B), it did contribute to COX-2 expression in FDC-like cells (Figure 2C). B cells do not express COX-2 protein [11]. The enhancing effects of B cells on COX-2 expression were dose-dependent and further strengthened by Akt knockdown. For instance, the addition of 30 × 10^5^ B cells resulted in a 4.2-fold increase in COX-2 expression, and Akt knockdown further elevated COX-2 levels by 2-fold (Figure 2D). The B cell subset that induced COX-2 in FDC-like cells was confirmed as activated B cells, as reported in our previous research [11]. In line with the results obtained with IL-1β, COX-1 expression was not affected by either B cell addition or Akt knockdown (Figure 2C).

### 2.3. Akt Is Not Involved in IL-4-Mediated Inhibition of COX-2 Induction in FDC-like Cells

Next, to determine the role of Akt in the inhibition of COX-2 expression and PG production in FDC-like cells, we first examined the effect of IL-4 on IL-1β-driven COX-2 expression based on the previous activity of this cytokine. IL-4 exhibits inhibitory effects on COX-2 induction with LPS. Cells were pre-incubated with IL-4 for 24 h, followed by further culture in the presence or absence of IL-1β for 4 h. Even though graded doses of IL-4 alone did not significantly modulate COX-2 expression, they inhibited IL-1β-stimulated COX-2 expression in a dose-dependent manner. IL-4 exhibited an inhibitory effect from 10 U/mL concentration and gave rise to an 80% reduction in IL-1β-induced COX-2 expression at 1000 U/mL concentration (Figure 3A). This inhibitory effect of IL-4 was confirmed in the actual production of 6-keto-PGF_1α_. As shown in Figure 3B, the dose-dependent reduction in PG recapitulated that of COX-2 protein expression. For example, IL-4 treatment at 1000 U/mL resulted in an 80% reduction in 6-keto-PGF_1α_ production. The effect of IL-4 on IL-1β-induced COX-2 expression seemed specific to this cytokine since IL-10, either alone or in combination with IL-4, did not further modulate COX-2 expression levels (Figure 3C). The inhibitory effect of IL-4 was specific to COX-2 and was not observed in the expression levels of COX-1.

Confirmation of the suppressive activity of IL-4 in COX-2 induction and PG production led us to analyze whether Akt was involved in this process. First, the effect of IL-4 on Akt activity was examined by measuring Akt phosphorylation degree. In time-course experiments, Akt phosphorylation levels were markedly elevated at the early time points, 15 and 30 min after IL-4 stimulation, and then returned to the background levels (Figure 4A). Similarly, in a dose-response experiment, IL-4 exhibited the augmenting effect on Akt phosphorylation from 10 U/mL concentration with the optimal effect at 100 or 1000 U/mL (Figure 4B). This result prompted us to explore the plausible role of Akt in COX-2 inhibition in IL-4-pretreated cells by comparing COX-2 induction levels after IL-1β stimulation between control and Akt-knockdowned cells. Akt knockdown was successfully carried out, as indicated by the average 88% (±5.2, *n* = 3) reduction in Akt protein level (Figure 4C). Akt siRNA transfection did not give rise to modulation of COX-2 expression in unstimulated cells. IL-1β increased COX-2 induction in both control and Akt-knockdowned cells with 50% further induction in the latter cells, as in line with the results in Figure 1A. IL-4 pretreatment almost completely prevented IL-1β-stimulated COX-2 induction in control siRNA-treated cells and resulted in 75% inhibition of COX-2 induction in cells treated with Akt siRNA, indicating the inhibitory effect of IL-4 was viable in Akt-knockdowned cells. Based on these data, we conclude that Akt, even though it is activated by IL-4, is not involved in IL-4-mediated inhibition of COX-2 induction with IL-1β in FDC-like cells.

### 2.4. STAT6 Is Necessary for IL-4 to Suppress the IL-1β-Driven COX-2 Expression

As we concluded that Akt was not the mediator of IL-4 suppression, we tested the possible contribution of the STAT molecules instead. In our previous investigation, IL-4 depended on STAT6 in its suppressive effect on PG production in FDC-like cells stimulated with TGF-β [12]. Therefore, we examined the effect of siRNA silencing of STAT molecules on the inhibitory activity of IL-4 in COX-2 production stimulated with IL-1β this time. Knockdown of STAT proteins except STAT1 did not significantly modulate the background level expression of COX-2 as compared to controls (Figure 5). STAT1 silencing resulted in a marked increase in background COX-2 levels. Consistent with the previous results [12], IL-4 pretreatment failed to affect the expression levels of COX-2 in STAT6-knockdowned cells compared to the control. Interestingly, IL-1β-induced COX-2 expression levels were significantly modulated after STAT1 and STAT3 knockdown. STAT1 silencing led to further augmentation of COX-2, whereas STAT3 silencing resulted in a marked inhibition of COX-2 induction. The modulation of IL-1β-induced COX-2 levels was not observed in the silencing of STAT2, STAT4, STAT5, and STAT6. Taken together, these results indicate that the inhibition of IL-1β-driven COX-2 expression by IL-4 is mediated via STAT6 in FDC-like cells.

## 3. Discussion

Important novel findings of this study are (1) the augmented COX-2 expression and PG production in Akt-knockdowned FDC-like cells stimulated with IL-1β or cocultured with B cells, (2) the marked suppression of COX-2 expression and PG production by IL-4, (3) the dispensable role of Akt in the suppressive activity of IL-4, (4) the dependence of IL-4 on STAT6 in its inhibition of IL-1β-stimulated COX-2 induction, and (5) the opposite effects of STAT1 and STAT3 silencing on IL-1β-stimulated COX-2 induction.

The results of the potent impact of IL-1β on COX-2 expression and resultant PG production in FDC-like cells reported here and previously provide insights into the role of this cytokine in humoral immunity. Although IL-1β has been established as one of the canonical pro-inflammatory cytokines, its role in the humoral immune response is largely unknown. The importance of IL-1β in humoral immunity is recently recognized, as reviewed by Ritvo and Klatzmann [18]. They suggest that IL-1β may play regulatory roles at the culminating site of antibody generation, GC, by directly inducing IL-4 and IL-21 from follicular helper T cells. The cellular sources of IL-1β in the GC appear to be macrophages or dendritic cells [19,20]. Our evidence obtained with FDC-like cells implies that IL-1β contributes to humoral immunity by stimulating COX-2 expression in FDC. Inasmuch as IL-1β blocking agents are currently therapeutic options for autoimmune or immune-inflammatory diseases [21], in vitro or in vivo dissection is necessary in the future to understand the biological role of IL-1β in humoral immune responses.

An interesting finding of this study is that Akt knockdown furthered COX-2 augmentation triggered by B cells in addition to that by IL-1β. Considering that the cell consists of extremely numerous molecules that may contain both inhibitory and stimulating ones, the finding that coculture with B cells leads to COX-2 upregulation in FDC-like cells is intriguing. Although the identity of the molecules responsible for COX-2 upregulation is currently unknown, TNF-α might be a COX-2-inducing molecule released from B cells. We have previously demonstrated that activated B cells induce TNF-α and COX-2 expression in FDC-like cells [11]. IL-1β was not detected in the conditioned media of activated B cells. Further investigation is necessary to identify the B cell molecules inducing COX-2 expression in FDC-like cells. As an intracellular mechanism underlying the B cell-induced COX-2 expression, we examined the potential role of Akt kinase. The results of this study imply that Akt kinase inhibits B cell-stimulated COX-2 expression because of the augmented COX-2 expression in Akt-knockdowned cells. Akt also exerts the inhibitory effect on COX-2 expression induced by IL-1β (Figure 1) or PGE_2_ [17]. However, Akt appears not to affect COX-2 expression in unstimulated FDC-like cells. Akt knockdown itself does not give rise to an increase in background-level expression of COX-2. Although B cells, IL-1β, and PGE_2_ are negatively affected by Akt on their stimulation pathway to COX-2 expression, they display differential effects on Akt phosphorylation. Both B cells and IL-1β do not modulate Akt phosphorylation, whereas PGE_2_ suppresses Akt activation. The significance of this differential effect remains to be investigated. The inhibitory effect of Akt on COX-2 expression in FDC-like cells is specific to these three stimuli because PGF_2α_-induced COX-2 expression is not affected by Akt knockdown [17]. In our attempt to investigate the cellular specificity of Akt knockdown, the inhibitory activity of Akt was also observed in 1064SK skin fibroblast cells (Appendix A). The significance of this result is currently unclear; however, it may indicate the common cellular origin of FDC and fibroblast hematopoietic mesenchymal cells [1,22,23]. Taken together, we maintain that Akt plays a negative role in COX-2 expression by indirectly acting on the induction pathways triggered by B cells, IL-1β, and PGE_2_. The direct target of Akt remains to be elucidated in future investigations.

The investigation of IL-4 in this study was instigated by our finding of the inhibitory activity of IL-4 in COX-2 protein expression and PG production by FDC-like cells. The effect of IL-4 on IL-1β-induced COX-2 has not been examined before. Since PGs are critical mediators of inflammation [24,25,26], we aimed to understand the regulatory mechanisms of PG production in the immune inflammatory milieu. IL-4 exhibited the inhibitory effect on COX-2 expression induced by IL-1β in the current study. As IL-4 inhibits COX-2 expression induced by other stimuli, including LPS, TGF-β, TNF-α, and IFN-γ, as reported by ours previously [12], this cytokine appears to play inhibitory roles in COX-2 induction irrespective of stimulus types. As a potential intracellular molecule conveying IL-4 signal to COX-2 inhibition, we examined Akt and found a dispensable function of this protein. Although the incomplete suppression of COX-2 induction by IL-4 in Akt-knockdowned cells does not exclude the possibility of a partial contribution of Akt, we maintain that this failure of IL-4 may have been caused probably because the inhibitory effect of Akt on COX-2 induction was mitigated after siRNA transfection. Instead of Akt, STAT6 was required for IL-4 to exhibit inhibitory activity on COX-2 induction. IL-4 belongs to the hematopoietin cytokine family, where signal transduction is achieved via the STAT pathway [27,28]. The dependence of IL-4 on STAT6 is in line with our previous findings [12]. Supportive of our finding and implying a critical role for IL-4 in the GC, Duan et al. have recently demonstrated that IL-4 plays an important role in the regulation of FDC-B cell interactions in the GC [29].

Another interesting finding of the current study is the regulatory effects of STAT1 and STAT3 proteins on IL-1β-stimulated COX-2 induction. Although the intracellular molecular pathways from IL-1β receptor to COX-2 expression are poorly known, the specific involvement of STAT1 and STAT3 in COX-2 induction is intriguing. STAT1 and STAT3 knockdown led to a further augmentation and a marked suppression of the stimulating effect of IL-1β, respectively. STAT1 appears to inhibit IL-1β-stimulated COX-2 induction, while STAT3 seems to support it. The detailed molecular mechanisms underlying these phenomena will be elucidated by our future studies.

In conclusion, the current data suggest that IL-4 and Akt are important exogenous and endogenous molecules suppressing PG production in FDC, and targeting these molecules may be an option in the treatment of immune-inflammatory disorders. Figure 6 shows our working model for the regulation of PG production in FDC based on the current experimental results. Indeed, STAT6 inhibitors or IL-4 antagonists are currently considered therapeutic options for immunopathologic diseases [30,31,32,33]. Considering the important roles that Akt plays in many physiological conditions, the effects of Akt inhibitors should be carefully analyzed when they are used as therapeutics in cancer, infectious diseases, and inflammation [34,35,36].

## 4. Material and Methods

### 4.1. Reagents

Immunoblotting to detect COX-2 (#12282S), total Akt (#9272) and phosphorylated Akt (Ser473, #9271), STAT1 (#9172), STAT2 (#4594), STAT3 (#9132), STAT4 (#2653), STAT5 (#9363), and STAT6 (#9362) proteins was carried out using antibodies from Cell Signaling Technology, Inc. (Danvers, MA, USA). Anti-COX-1 antibody (#sc-19998) was purchased from Santa Cruz Biotechnology, Inc. (Dallas, TX, USA). Anti-β-actin antibody (#5441) was obtained from Sigma-Aldrich (St. Louis, MO, USA). Horseradish peroxidase-conjugated anti-mouse IgG (#1031-05) and anti-rabbit IgG (#6721) antibodies were purchased from Southern Biotech and Abcam (Cambridge, UK), respectively. IL-1β (#201-LB) was a product of R&D Systems (Minneapolis, MN, USA), and recombinant IL-4 and IL-10 were prepared in our laboratory [27]. PGE_2_ (#14010) and EIA kits to measure PGI_2_ (#515211) were purchased from Cayman Chemicals (Ann Arbor, MI, USA). FITC- or PE-conjugated anti-CD3 (clone HIT3α), CD20 (clone 2H7), and anti-CD38 (clone HIT2) antibodies were products of BD Biosciences (San Jose, CA, USA). Akt siRNA (#6211) was obtained from Cell Signaling Technology, Inc. (Danvers, MA, USA). siRNAs against STAT1 (4390824-s277), STAT2 (4392420-s13530), STAT3 (4390824-s743), STAT4 (4392420-s13531), STAT5 (4392420-s13537), STAT6 (4390824-s13542), and control siRNA (Neg-siRNA#2, sequence not disclosed by the manufacturer) were purchased from Ambion Inc., Austin, TX, USA.

### 4.2. Isolation of FDC-like Cells and B Cells

This study was approved by the ethics committee of Kangwon National University (2019-09-004-002). Human tonsils obtained after routine tonsillectomy were treated as described previously to prepare B cells and FDC-like cells [8,37]. The purity of isolated FDC-like cells was evaluated using a FACSCalibur (Becton Dickinson, Franklin Lakes, NJ, USA) after staining with an anti-PGIS antibody (clone 3C8). FDC-like cells obtained from different donors did not fail to display increased COX-2 expression in response to IL-1β stimulation (Appendix A), indicating the consistency between different FDC-like cell batches. Mononuclear cells were separated by the method of discontinuous centrifugation using Ficoll-Hypague, which were further subfractionated to B cells after depletion of T cells with rosetting with sheep blood cells [38]. B cell purity was verified using a FACSCalibur. The culture medium for FDC-like cells and B cells was RPMI-1640 (Gibco), containing 10% fetal calf serum (Hyclone, Logan, UT, USA), 100 U/mL penicillin G (Sigma), 100 µg/mL streptomycin (Invitrogen, Waltham, MA, USA), and 2 mM L-glutamine (Life Technologies, Carlsbad, CA, USA).

### 4.3. Enzyme Immunoassay to Measure PGI_2_ (6-keto-PGF_1α_)

The amounts of 6-keto-PGF_1α_ in the culture supernatants were measured with immunoassay kits by following the manufacturer’s protocols [39]. In brief, the diluted supernatants and standards were added to each well of the plate in triplicate with equal volumes of 6-keto-PGF_1α_ AChE tracer and 6-keto-PGF_1α_ antiserum. The plate was incubated at 4 °C for 18 h, followed by the development with Ellman’s reagent after washing. The absorbance of the plate was measured by reading at 405 nm, followed by the data analysis with GraphPad Prism 5.01 software.

### 4.4. siRNA Transfection

Transfection of FDC-like cells with Akt or control siRNA was carried out as described before [17]. FDC-like cells were subjected to siRNA transfection when the 60 mm plates were 70–80% full. Each siRNA (20 nM) and lipofectamine RNAiMAX (12 μL) for each plate were separately diluted in 400 μL serum-free medium without antibiotics, mixed together, and then incubated for 5 min at RT. The plate containing FDC-like cells was washed with serum-free medium, added with 1.6 mL serum-free medium, and then with the diluted mixture. The plates were incubated for 8 h at 37 °C, followed by the addition of a serum-free medium. After 48 h of additional incubation, cells were stimulated with IL-1β or B cells used for experiments. The degree of gene silencing was verified with immunoblotting.

### 4.5. Immunoblotting

Immunoblotting was carried out as described before [17]. In brief, cultured cells were washed with cold PBS and then lysed in Pro-prep cell extract solution (Intron Biotechnology) on ice, followed by centrifugation of lysates at 15,000× *g* for 10 min at 4 °C to remove any insoluble debris. Protein quantification was carried out using a BCA protein assay kit (Thermo Scientific, Waltham, MA, USA). After a routine SDS-PAGE, proteins were transferred onto polyvinylidene difluoride membranes. Blocking of the membranes was performed with 3% nonfat-dried milk for 1 h. After 3 times washing, the membranes were treated with specific antibodies. Visualization of the detected proteins was carried out using a chemiluminescent solution (Advansta, San Jose, CA, USA) and X-ray films [40].

### 4.6. Statistical Analyses

Processing of experimental results and statistical analyses were carried out using GraphPad Prism 5.01 software to present the mean ± SEM of at least three independent and reproducible experiments. The statistical significance of differences was determined using Student’s *t*-test. *p* < 0.05 was considered significant.

## Figures and Tables

**Figure 1 ijms-24-16692-f001:**
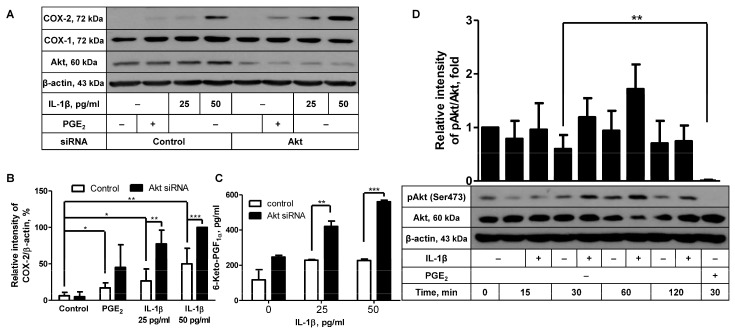
The effects of Akt knockdown on IL-1β-induced COX-2 expression. (**A**) The effect of Akt knockdown on IL-1β-induced COX-2 expression was analyzed with immunoblotting with FDC-like cells cultured with the suboptimal doses of IL-1β for 4 h. PGE_2_ (10 µM) was included in the culture to confirm our previous results for the effect of Akt knockdown. (**B**) Statistical analysis was carried out with the results of (**A**) and two more independent experiments. (**C**) The culture of FDC-like cells was maintained for 24 h, followed by the harvest of supernatants to measure 6-keto-PGF_1α_ with EIA. (**D**) FDC-like cells were harvested at the end of indicated time points after culture in the presence or absence of IL-1β, followed by immunoblotting to measure expression levels of total and phosphorylated Akt proteins. PGE_2_ was included in the culture as a positive control for the downregulation of Akt phosphorylation. Representative immunoblots and statistical analysis data (mean ± SEM) from three independent experiments are shown. Statistical significance was analyzed with Student’s *t*-test by comparing the values with the control (* *p* < 0.05, ** *p* < 0.01, *** *p* < 0.001).

**Figure 2 ijms-24-16692-f002:**
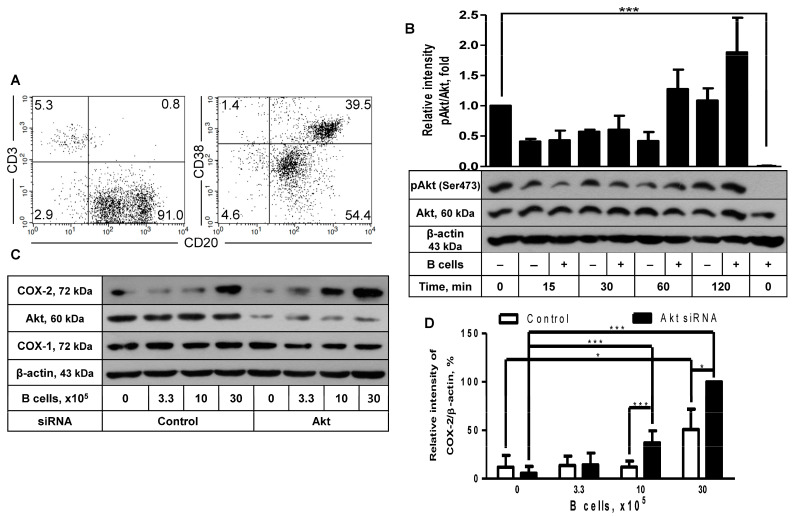
The effect of Akt knockdown on COX-2 expression in FDC-like cells cocultured with B cells. (**A**) B cells were freshly isolated from the tonsils. The purity of separated cells was measured with flow cytometric analysis, as shown in a representative result. Numbers are cell percentages in each quadrant. (**B**) FDC-like cells (3 × 10^5^ cells in a 60 mm dish) were cultured in the presence or absence of tonsillar B cells (3 × 10^6^ cells) for the indicated time periods, followed by immunoblotting to measure expression levels of total and phosphorylated Akt proteins. (**C**,**D**) The effect of Akt knockdown on COX-2 induction was analyzed with immunoblotting after culturing FDC-like cells with the indicated numbers of B cells together for 24 h. Representative immunoblots and statistical analysis data (mean ± SEM) from three independent experiments are shown. The successful knockdown of Akt was verified with immunoblotting. Statistical significance was analyzed with Student’s *t*-test by comparing the values with the control (* *p* < 0.05, *** *p* < 0.001).

**Figure 3 ijms-24-16692-f003:**
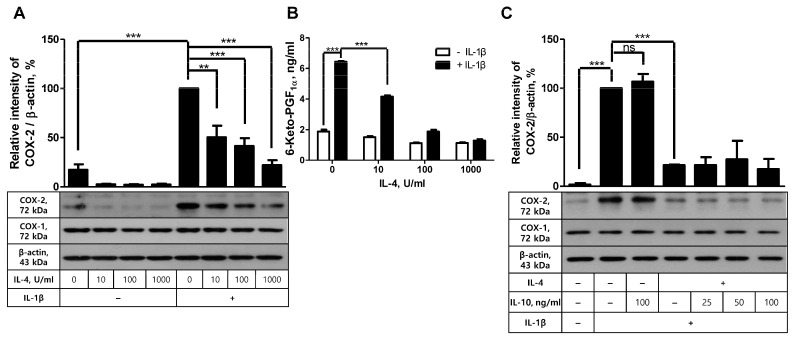
The suppressive effects of IL-4 on COX-2 induction and PG production. (**A**) FDC-like cells were pretreated with the indicated doses of IL-4 for 24 h, followed by further culture in the presence or absence of IL-1β for 4 h. The expression levels of COX-2 were measured with immunoblotting. (**B**) The culture of FDC-like cells was maintained for 24 h, followed by EIA with the supernatants to measure 6-keto-PGF_1α_. (**C**) FDC-like cells were pretreated with the indicated doses of IL-10 or IL-4 (100 U/mL) for 24 h, followed by further culture with IL-1β for 4 h. The expression levels of COX-2 were measured with immunoblotting. Representative immunoblots and statistical analysis data (mean ± SEM) from three independent experiments are shown. The successful knockdown of Akt was verified with immunoblotting. Statistical significance was analyzed with Student’s *t*-test by comparing the values with the control (** *p* < 0.01; *** *p* < 0.001; ns, nonsignificant).

**Figure 4 ijms-24-16692-f004:**
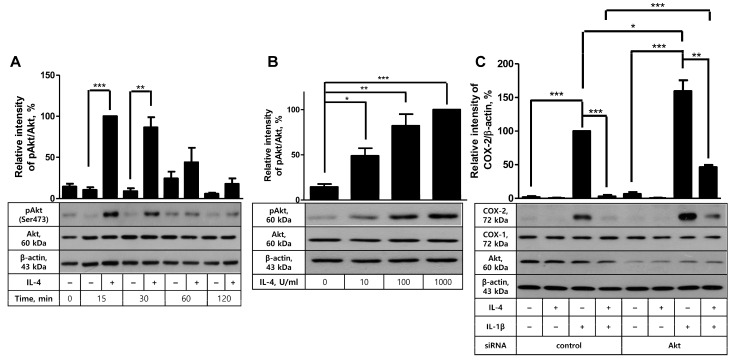
The effects of IL-4 on Akt phosphorylation and Akt knockdown on the suppressive activity of IL-4. (**A**) FDC-like cells (3 × 10^5^ cells in a 60 mm dish) were cultured in the presence or absence of IL-4 (100 U/mL) for the indicated time periods, followed by immunoblotting to measure expression levels of total and phosphorylated Akt proteins. (**B**) FDC-like cells were cultured with the indicated doses of IL-4 for 30 min, followed by immunoblotting to measure expression levels of total and phosphorylated Akt proteins. (**C**) The effect of Akt knockdown on the suppressive activity of IL-4 was analyzed with immunoblotting after culturing FDC-like cells with IL-4 (100 U/mL) for 24 h and then with IL-1β (25 pg/mL) for 4 h. Representative immunoblots and statistical analysis data (mean ± SEM) of three independent experiments are shown. Statistical significance was analyzed with Student’s *t*-test by comparing the values with the control (* *p* < 0.05, ** *p* < 0.01, *** *p* < 0.001).

**Figure 5 ijms-24-16692-f005:**
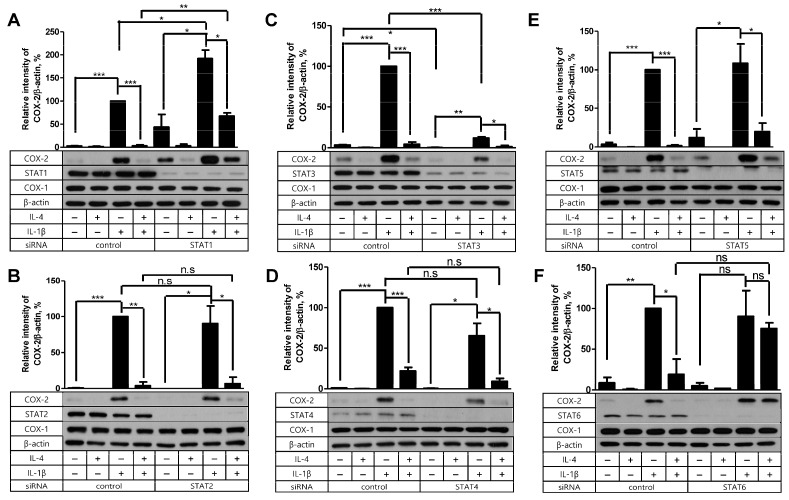
The effect of STAT protein knockdown on the suppressive activity of IL-4. (**A**–**F**) Indicated STAT proteins or control siRNA-transfected FDC-like cells were cultured in the presence or absence of IL-4 (100 U/mL) for 24 h and then with IL-1β (25 pg/mL) for 4 h. Expression levels of indicated proteins were measured with immunoblotting. Representative immunoblots and statistical analysis data (mean ± SEM) from three independent experiments are shown. The successful knockdown of Akt was verified with immunoblotting. Molecular weights of STATs: STAT1 (91 kDa), STAT2 (113 kDa), STAT3 (86 kDa), STAT4 (81 kDa), STAT5 (90 kDa), and STAT6 (110 kDa). Statistical significance was analyzed with Student’s *t*-test by comparing with the control (* *p* < 0.05; ** *p* < 0.01; *** *p* < 0.001; ns, nonsignificant).

**Figure 6 ijms-24-16692-f006:**
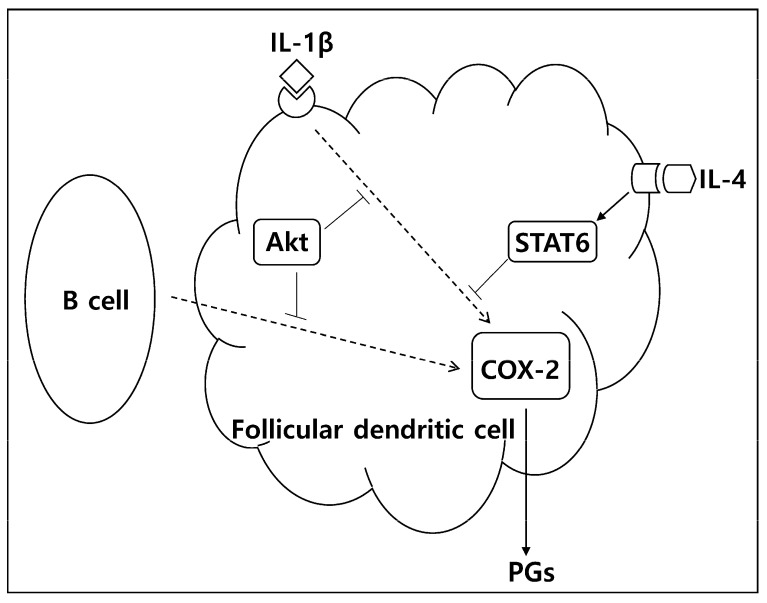
A proposed model for the regulation of PG production in FDC. PGs are produced in FDC during cellular interactions with B cells or in response to inflammatory cytokines, including IL-1β. The upstream enzyme of PGs, COX-2, is increased in FDC after B cell stimulation and IL-1β triggering. Akt plays inhibitory roles during these processes by restraining COX-2 expression. IL-4 also suppresses PG production by repressing COX-2 expression in a STAT6-dependent manner.

## Data Availability

The data used in this research are available upon request from the corresponding author.

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
