# Peer review of "Akt, IL-4, and STAT Proteins Play Distinct Roles in Prostaglandin Production in Human Follicular Dendritic Cell-like Cells"

_ijms, 2023, doi:10.3390/ijms242316692_

Round 1

Reviewer 1 Report (New Reviewer)

Comments and Suggestions for Authors

This study explored the role of AKT in PG production in FDC-like cells and found that AKT plays a negative role in IL-1b-induced PG production.  Overall, this study was well designed and the interpretation is clear and sufficient.  Please see below my minor comments.

1, Figure 1, how many Akt siRNAs were designed and validated?  It would be better to show more than one siRNA to exclude the potential off-target effect.  Also, the authors should always show phospho-AKT levels in their KD conditions since there are still AKT expression.  Quantification of panel D is required since total AKT levels were significantly affected.

2, Figure 2, labels are missing in panel A.  Quantification of panel B is required.  Phospho-AKT is required in panel C.

3, Figure 3, which concentration of IL-1b was used should be marked in the figures.

4, Figure 4, phospho-AKT needs to be evaluated in panel C. 

5, Figure 5, quantifications are missing in panel B, D and F.

Comments on the Quality of English Language

Overall, the interpretation is logical and clear, no obvious grammar mistakes was observed.  

Author Response

We appreciate the highly constructive comments from the reviewer #1 and respond to them as follows.

Point-by-point reply to the reviewer #1:

Reviewer #1: This study explored the role of AKT in PG production in FDC-like cells and found that AKT plays a negative role in IL-1b-induced PG production.  Overall, this study was well designed and the interpretation is clear and sufficient.  Please see below my minor comments.

1, Figure 1, how many Akt siRNAs were designed and validated?  It would be better to show more than one siRNA to exclude the potential off-target effect.  Also, the authors should always show phospho-AKT levels in their KD conditions since there are still AKT expression.  Quantification of panel D is required since total AKT levels were significantly affected.

- Akt siRNA was not designed in our laboratory. We purchased the validated product from Cell Signaling Technology. The missing information was inserted into 4.1.

- We measured Akt in Figure 1A to evaluate the knockdown efficacy. Instigated by the reviewer’s comment, however, we repeated the experiments measuring phosphorylated Akt levels this time (Supplementary Figure S1A).

- Figure 1D was revised with the quantification data. In the revised figures or legend, we indicated the molecular weights of the proteins detected.

2, Figure 2, labels are missing in panel A.  Quantification of panel B is required.  Phospho-AKT is required in panel C.

- There are antibody labels in Figure 2A; CD3, CD20, and CD38. We enhanced the image clarity of Figure 2A.

- Figure 2B was revised with the quantification data.

- Phospho-Akt levels were detected in Supplementary Figure S1 instead.

3, Figure 3, which concentration of IL-1b was used should be marked in the figures.

We now indicated the concentration of IL-1β in Figure 4.

4, Figure 4, phospho-AKT needs to be evaluated in panel C.

We measured Akt in Figure 4C to evaluate the knockdown efficacy. Instigated by the reviewer’s comment, however, we repeated the experiments measuring phosphorylated Akt levels this time (Supplementary Figure S1B).

5, Figure 5, quantifications are missing in panel B, D and F.

Figure 5 was revised with the quantification data.

Reviewer 2 Report (New Reviewer)

Comments and Suggestions for Authors

The paper describes the impact of the signalling molecule AKT on COX-2 protein expression induce by IL-1b or inhibited by IL-4 in in vitro derived follicular dendritic cells (FDC). The paper identified the complex relationship between AKT, STAT1 and STAT6 on IL-1b induction and IL-4 inhibition of cox 2. The strength of the paper was a simple model with detailed and consistent approach of immunoblotting to investigate protein expression.

A statement to explain why protein expression (immunoblotting) was the main technique used without any gene expression studies in the introduction is needed.

This work continues to explore in an incremental manner the constituents required for regulation of prostaglandin expression in a FDC model. Whilst this model may be commonly used with literature citing the model common in the archives, the reference used in the article under review did not adequately inform the readers that the actual origins and characterisation of the cells were originally described in a J Immunol paper by Ed Clark et al., in 1992. There was no characterization or QC of the cell preps to show consistency between preparations from different tonsils. Tonsils are often inflamed when removed and this may impact the derived cells.

 Given that tonsils are inherently different, it is likely that each tonsil provided cells that were also different in their activation status, viability, numbers etc. There is no acknowledgement of this variability.

The figure legends need to reflect the data. In Fig 1B, the legend describes statistical analysis carried out on the results of A. Yet the results of A are a representative of one of 3. In this case Fig 1B should not have error bars. If Fig1B represents data from all 3 experiments, displaying the individual results on the bar graphs would allow assessment of the data variability.

Similar comments can be made for all figure legends. For example in Fig 2A, the dotplots represent the purity but also the division of the B cell subsets. In Fig 3A what do the error bars represent on the graph if the graph represents the single data point from the immunoblot?

The impact of B cells on the induction of cox-2 suggested that activated B cells were the active population. This data needs to be shown. The data in Figure 2 suggests 3.3 and 10 x10e5 B cells did not significantly impact cox2 although the higher number did. Is this due to using a heterogeneous sample of activated and un-activated B cells?

Only in vitro derived FDCs were used and so it is unknown whether the results are generally applicable to AKT function or whether this is just in FDC. Is there a human cell line that can be used to determine if AKT function is regulated in this manner by cells other than FDC? How can this be shown to be relevant physiologically?

Sixteen out of 38 references appear to be self citations indicating that the authors publish often in this field and it is a field possible restricted by the model. Are there other publications to justify using this model?

The hypothesis and conclusions could be written with more clarity connecting the figures – perhaps in a schema. The title reflects the focus on AKT, IL4 and STAT proteins however the introduction and conclusions indicate that the focus is on understanding AKT induction and inhibition of PG. Each figure centres on a different set of signaling components of COX2 protein expression to build a picture.

Comments on the Quality of English Language

The paper is reasonably well written with a few grammatical errors that would be identified in proof reading stage.

Author Response

We appreciate the highly constructive comments from the reviewer #2 and respond to them as follows.

Point-by-point reply to the reviewer #2:

Reviewer #2: The paper describes the impact of the signalling molecule AKT on COX-2 protein expression induce by IL-1b or inhibited by IL-4 in in vitro derived follicular dendritic cells (FDC). The paper identified the complex relationship between AKT, STAT1 and STAT6 on IL-1b induction and IL-4 inhibition of cox 2. The strength of the paper was a simple model with detailed and consistent approach of immunoblotting to investigate protein expression.

A statement to explain why protein expression (immunoblotting) was the main technique used without any gene expression studies in the introduction is needed.

The merits for using immunoblotting in this study are described now in Introduction.

This work continues to explore in an incremental manner the constituents required for regulation of prostaglandin expression in a FDC model. Whilst this model may be commonly used with literature citing the model common in the archives, the reference used in the article under review did not adequately inform the readers that the actual origins and characterisation of the cells were originally described in a J Immunol paper by Ed Clark et al., in 1992. There was no characterization or QC of the cell preps to show consistency between preparations from different tonsils. Tonsils are often inflamed when removed and this may impact the derived cells. Given that tonsils are inherently different, it is likely that each tonsil provided cells that were also different in their activation status, viability, numbers etc. There is no acknowledgement of this variability.

- According to the reviewer’s comment, we now replace the reference #11 with Edward Clark et al.

- As a way of showing QC of the cell preps, we provide additional data (Supplementary Figure S2).

The figure legends need to reflect the data. In Fig 1B, the legend describes statistical analysis carried out on the results of A. Yet the results of A are a representative of one of 3. In this case Fig 1B should not have error bars. If Fig1B represents data from all 3 experiments, displaying the individual results on the bar graphs would allow assessment of the data variability.

The legend of Figure 1B was corrected according to the reviewer’s comment.

Similar comments can be made for all figure legends. For example in Fig 2A, the dot plots represent the purity but also the division of the B cell subsets. In Fig 3A what do the error bars represent on the graph if the graph represents the single data point from the immunoblot?

As described in the legends of Figure 2 and 3, the immunoblots are representative of three independent experiments and the statistical analyses are performed with the results of representative and two more experiments.

The impact of B cells on the induction of cox-2 suggested that activated B cells were the active population. This data needs to be shown. The data in Figure 2 suggests 3.3 and 10 x10e5 B cells did not significantly impact cox2 although the higher number did. Is this due to using a heterogeneous sample of activated and un-activated B cells?

The number of B cells required to induce COX-2 significantly in FDC-like cells was pre-determined as reported in reference #14. Since Akt knockdown further augmented COX-2 expression induced by B cells, we co-cultured with the suboptimal numbers of B cells (3.3, 10, 30 â…¹ 105 cells) to demonstrate the dose-responsive results after Akt knockdown.

Only in vitro derived FDCs were used and so it is unknown whether the results are generally applicable to AKT function or whether this is just in FDC. Is there a human cell line that can be used to determine if AKT function is regulated in this manner by cells other than FDC? How can this be shown to be relevant physiologically?

We examined the effect of Akt knockdown with 1064SK human cell line and obtained similar results (Figure S2). The physiological significance of our finding is described in 2.1.

Sixteen out of 38 references appear to be self-citations indicating that the authors publish often in this field and it is a field possible restricted by the model. Are there other publications to justify using this model?

The usefulness of this model as a mean to study human FDC is confirmed by the publication of other laboratories. Followings are some examples.

  1. FDC like cell line HK with IL-2, IL-4 and OX40 ligand supports the growth of ATLL cells, Blood (2013), 122, 4319.
  2. Follicular dendritic cell (FDC)-induced microrna-mediated up-regulation of PRDM1 and down-regulation of BCL6 in germinal center B lymphocytes: a potential mechanism for B-cell differentiation, Blood (2009), 114, 4588.
  3. The prevention of spontaneous apoptosis of follicular lymphoma B cells by a follicular dendritic cell line: involvement of caspase-3, caspase-8 and c-FLIP, Hematologica (2008), 93, 1169.

The hypothesis and conclusions could be written with more clarity connecting the figures – perhaps in a schema. The title reflects the focus on AKT, IL4 and STAT proteins however the introduction and conclusions indicate that the focus is on understanding AKT induction and inhibition of PG. Each figure centres on a different set of signaling components of COX2 protein expression to build a picture.

According to the reviewer’s comment, we now provide Figure 6.

This manuscript is a resubmission of an earlier submission. The following is a list of the peer review reports and author responses from that submission.

Round 1

Reviewer 1 Report

Comments and Suggestions for Authors

I am thankful to the editor for providing me with this opportunity to review the manuscript. The study by Jeong & Choe is very well-designed and well-executed. I have a few minor suggestions and queries.

1. What's the rationale for taking 105 B cells in the co-culture?

2. Kindly provide the molecular weight of the proteins in the western blot images.

3. Kindly enhance the image clarity of the blots and flow cytometry plot.

4. In Statistics, what's the reason for calculating S.D., why not SEM? Are the events independent or not?

5. Kindly provide the product code for the antibodies used in the study.

6. kindly describe the methodology briefly in 4.3 -Enzyme immunoassay to measure PGl2

Author Response

We appreciate the highly constructive comments from the three reviewers and respond to them as follows.

Point-by-point reply to the reviewer #1:

Reviewer #1: I am thankful to the editor for providing me with this opportunity to review the manuscript. The study by Jeong & Choe is very well-designed and well-executed. I have a few minor suggestions and queries.

  1. What's the rationale for taking 105 B cells in the co-culture?

The number of B cells required to induce COX-2 in FDC-like cells (3 â…¹ 105 cells in a 60-mm dish) was pre-determined as reported in reference #14. Since Akt knockdown further augmented COX-2 expression induced by B cells, we co-cultured with the suboptimal numbers of B cells (3.3, 10, 30 â…¹ 105 cells) to demonstrate the dose-responsive results after Akt knockdown.

  1. Kindly provide the molecular weight of the proteins in the western blot images.

According to the reviewer’s comment, we now provide the molecular weights of the proteins in the western blot figures.

  1. Kindly enhance the image clarity of the blots and flow cytometry plot.

We replaced Figure 2A with the enhanced clarity images. We find that the clarity quality of immunoblot images is lowered during the conversion from ppt to pdf. We provide the MDPI office with additional ppt data containing the original image quality.

  1. In Statistics, what's the reason for calculating S.D., why not SEM? Are the events independent or not?

In response to the reviewer’s comment, we present all the data as the mean ± SEM. The mean and SEM were calculated with the results of at least three independent and reproducible experiments as described in 4.6 Statistical analyses.

  1. Kindly provide the product code for the antibodies used in the study.

We revised 4.1 Reagents to include the product codes for the used antibodies and other reagents.

  1. kindly describe the methodology briefly in 4.3 -Enzyme immunoassay to measure PGl2.

We revised the 4.3 section and briefly described the enzyme immunoassay procedure.

Reviewer 2 Report

Comments and Suggestions for Authors

In this study, the inhibitory role of Akt and IL-4 in the prostaglandin production by human follicular dendritic cell-like cells is investigated. The experiments appear to be well conducted, but it’s hard to appreciate the relevance of the results with respect to those already published by the same research group, and in the more general context of interaction between FDCs and B cells. In its present form, the paper is difficult to read and understand and, and in my opinion, an extensive revision is needed. In the Introduction paragraph, it is necessary to (i) organically summarize the previous knowledge on the specific research subject (already explored by the same team), and (ii) consistently indicate the objectives to be achieved with the proposed research. This information is scattered among the Results and Discussion sections, this making difficult to understand what’s new in the present paper.

Author Response

We appreciate the highly constructive comments from the three reviewers and respond to them as follows.

Point-by-point reply to the reviewer #2:

Reviewer #2: In this study, the inhibitory role of Akt and IL-4 in the prostaglandin production by human follicular dendritic cell-like cells is investigated. The experiments appear to be well conducted, but it’s hard to appreciate the relevance of the results with respect to those already published by the same research group, and in the more general context of interaction between FDCs and B cells. In its present form, the paper is difficult to read and understand and, and in my opinion, an extensive revision is needed. In the Introduction paragraph, it is necessary to (i) organically summarize the previous knowledge on the specific research subject (already explored by the same team), and (ii) consistently indicate the objectives to be achieved with the proposed research. This information is scattered among the Results and Discussion sections, this making difficult to understand what’s new in the present paper.

According to the reviewer’s suggestion, we revised Introduction to summarize our previous results on the induction and inhibition of COX-2. We also summarized our data on in vitro and in vivo function of PGs. The objectives of this study are presented in the last paragraph of Introduction. The novel findings of this study are summarized as 1) ~ 4) in the first paragraph of Discussion.

Reviewer 3 Report

Comments and Suggestions for Authors

1. The major comment for this article is that the novelty and significance of the study are very limited. Previous studies have already shown about the inhibitory roles of Akt and IL-4 in prostaglandin production in FDCs. The difference is just on the inducers: IL-1β in current study while PGE2, LPS and so on in previous studies. The present study only investigated the roles of Akt and IL-4 in vitro in FDC-like cells. It is hard to tell how the findings can relate to the physiological and/or disease conditions. More solid experiments and in vivo study are missing. 

2. Why it is important to use FDC and study PG production? It should be mentioned in the Introduction. 

3. There are different phosphorylation sites for Akt. Which was measured by the authors? The phosphorylation site and protein size should be indicated clearly in figures and/or Materials and methods. 

Author Response

We appreciate the highly constructive comments from the three reviewers and respond to them as follows.

Point-by-point reply to the reviewer #3:

  1. The major comment for this article is that the novelty and significance of the study are very limited. Previous studies have already shown about the inhibitory roles of Akt and IL-4 in prostaglandin production in FDCs. The difference is just on the inducers: IL-1β in current study while PGE2, LPS and so on in previous studies. The present study only investigated the roles of Akt and IL-4 in vitro in FDC-like cells. It is hard to tell how the findings can relate to the physiological and/or disease conditions. More solid experiments and in vivo study are missing. 

IL-1β has recently been recognized as an important regulator of humoral immune responses (Ref. #28). IL-1β is the most potent inducer of COX-2 in FDC-like cells in our laboratory. Since we have demonstrated FDC as the major cell expressing COX-2 in situ (Ref. #14), we aimed in this study to investigate the roles for Akt in IL-1β-induced PG production and IL-4-mediated inhibition of PG production in FDC-like cells. Furthermore, the role for Akt was also newly explored in the COX-2 induction after co-culture with B cells.

  1. Why it is important to use FDC and study PG production? It should be mentioned in the Introduction. 

In response to the reviewer’s highly constructive comment, we revised Introduction and described why it is important to study FDC and PG.

  1. There are different phosphorylation sites for Akt. Which was measured by the authors? The phosphorylation site and protein size should be indicated clearly in figures and/or Materials and methods. 

According to the reviewer’s comment, we now provide the phosphorylation site and molecular weights of Akt proteins in the figures.

Round 2

Reviewer 2 Report

Comments and Suggestions for Authors

Overall, the authors responded adequately to the comments: the article was significantly improved and is ready for publication.

Reviewer 3 Report

Comments and Suggestions for Authors

The authors have addressed my second and third comments. But they did not fully address my first comment which is my major concern about novelty and significance of current study.